# Effect of Sodium-Glucose Co-Transporter 2 Inhibitors Combined with Metformin on Pancreatic Cancer Cell Lines

**DOI:** 10.3390/ijms25189932

**Published:** 2024-09-14

**Authors:** André Cristovão, Nelson Andrade, Fátima Martel, Cláudia Silva

**Affiliations:** 1Unit of Biochemistry, Department of Biomedicine, Faculty of Medicine of Porto, University of Porto, 4200-319 Porto, Portugal; up202203829@edu.med.up.pt (A.C.); nandrade@med.up.pt (N.A.); claudiasilva@med.up.pt (C.S.); 2REQUIMTE/LAQV, Department of Chemical Sciences, Faculty of Pharmacy, University of Porto, 4200-135 Porto, Portugal; 3Instituto de Investigação e Inovação em Saúde (I3S), University of Porto, 4200-135 Porto, Portugal

**Keywords:** Panc-1 and AsPC-1 pancreatic cancer cell lines, SGLT2 inhibitors, metformin, PI3K and JNK intracellular signaling pathways

## Abstract

Pancreatic cancer (PC) is the ninth-leading cause of cancer-related deaths worldwide. Diabetic patients have an increased risk and mortality rates for PC. Sodium-glucose co-transporter 2 (SGLT2) inhibitors and metformin (Met) are widely used anti-diabetic medications. Both Met and SGLT2 inhibitors have anticancer properties in PC, but nothing is known concerning their combined effect. So, we investigated the in vitro effect of SGLT2 inhibitors combined with Met. Canagliflozin and dapagliflozin possessed cytotoxic, antiproliferative, and pro-apoptotic properties in the tested PC cell lines. In PANC-1 cells, the antimigratory and pro-apoptotic effects were enhanced when dapagliflozin was combined with Met, and G1 cell cycle arrest was enhanced when dapagliflozin or canagliflozin was combined with Met. In AsPC-1 cells, the cytotoxic effect and the G1 cell cycle arrest were enhanced when canagliflozin and dapagliflozin, respectively, were combined with Met. Only the cytotoxic effects of SGLT2 inhibitors, but not the combination treatments, involved PI3K and JNK-dependent pathways in AsPC-1 cells. In conclusion, combination treatments increased the anticancer effects in a cell type-dependent way in the two investigated cell lines. Additionally, the cytotoxic effect of SGLT2 inhibitors was dependent on the PI3K and JNK pathways in AsPC-1 cells, but Met appears to act via a distinct mechanism.

## 1. Introduction

Pancreatic cancer (PC) affects thousands of people every year, and in 2020, it was the 12th most common type of cancer, affecting 495,773 people, and the cancer with the 7th-highest mortality rate, responsible for the deaths of 466,003 people [1,2]. Future trends for this cancer type are not encouraging and it is expected that, by the year 2040, the number of people affected by PC will rise to 844,000 and the number of deaths to 801,000 [1]. This increasing trend is related to the fact that several risk factors that increase the predisposition to developing this neoplasm, including obesity and diabetes, have an increasing incidence worldwide [3,4,5,6,7]. It is also known that survival rates 5 years after diagnosis of PC are extremely low, at around 9% in 2019 in the USA and Europe. One of the reasons for the extremely low survival rates is because PC is often diagnosed at a very advanced stage [8].

Metformin is a first-line drug (alone or in combination) for type 2 diabetes (T2DM) therapy and constitutes the most commonly prescribed drug for this condition worldwide [9,10]. Its main mechanism of action consists of inhibition of the production of glucose by the liver [9,10]. However, in addition to this mechanism, it is known that this drug can have other effects on the body [10,11,12]. Importantly, several epidemiological and preclinical studies have presented consistent and robust evidence that metformin has beneficial effects on PC, such as reducing the risk of developing this disease and improving survival in early-stage patients [13].

Gliflozins, a novel class of oral anti-diabetic drugs, act by selectively inhibiting the type 2 sodium-glucose co-transporter (SGLT2), blocking glucose reabsorption in the S1 and S2 segments of the nephron, with a resulting greater glucose excretion in the urine [14]. SGLT2 inhibitors such as dapagliflozin and canagliflozin have a high affinity for SGLT2 and a low affinity for SGLT1, which contributes to their renal selectivity [14,15]. Because of their favorable effects on cardiovascular risk and renal disease progression, they are considered to be among the first-choice drugs in the case of metformin failure. Importantly, the literature describes overexpression of SGLT2 in various types of cancer, including PC [16,17], and some studies concluded that SGLT2 inhibitors may have a beneficial effect in PC therapy [16,17]. Thus, the use of SGLT2 inhibitors has emerged as a possible novel anticancer therapy for PC.

Developing new anticancer drugs is expensive, takes a long time, and has a high risk of failure. Because of this, the pharmaceutical industry has sought alternative methods to speed up and simplify the process. In this sense, the concept of drug repositioning or reprofiling has emerged. This is a method that uses approved drugs for diseases that have not previously been used. This strategy can provide new treatment options for cancer patients, saving time, energy, and money, and allow direct entry into clinical trials, since the drugs have already undergone the establishment of toxicity and safety profiles [18,19,20].

Many T2DM patients do not achieve glycemic control with metformin alone and eventually require a combination therapy with other agents such as SGLT2 inhibitors [21,22,23]. Because nothing is known concerning the effect of the combination of metformin and SGLT2 inhibitors on PC, we decided to evaluate the effect of SGLT2 inhibitors (canagliflozin and dapagliflozin), separately and in combination with metformin, on several cancer-related characteristics of the PC cell lines AsPC-1 and PANC-1.

## 2. Results

### 2.1. Effects of Canagliflozin (Can), Dapagliflozin (Dap), Metformin (Met), or Cisplatin (Cis) on Cell Viability and Cell Proliferation

In the first series of experiments, three different cell lines were used: two pancreatic tumor cell lines (AsPC-1 and PANC-1) and a normal skin fibroblast cell line (HDFa). The HDFa cell line was used to assess the selectivity of the drugs in relation to the cancer cell lines.

The effect of different concentrations of Can, Dap, Met, and Cis on the viability of the AsPC-1, PANC-1, and HDFa cell lines is shown in Figure 1a, Figure 1b, and Figure 1c, respectively. In the AsPC-1 cell line, there is a significant, concentration-dependent reduction in viability after treatment with Can (10 μM and 100 μM), Dap (1 μM and 100 μM), Met (5 μM, 50 μM, and 500 μM), and Cis (0.5 mM and 1 mM) (Figure 1a). In the PANC-1 cell line, there is also a significant, concentration-dependent reduction in cell viability after treatments with Can (10 μM and 100 μM), Met (500 μM), and Cis (0.5 mM and 1 mM) (Figure 1b). Finally, in the HDFa cell line, there is a significant reduction in cell viability after treatments with Can (100 μM) and Cis (0.5 mM and 1 mM), and, in contrast, a significant increase in cell viability with Met 5 μM (Figure 1c).

Figure 1d–f show the effect of different concentrations of Can, Dap, Met, and Cis on the proliferation rates of the three cell lines.

In the AsPC-1 cell line, there was a significant concentration-dependent decrease in cell proliferation after treatment with Can (10 μM and 100 μM), Dap (100 μM), and Cis (0.5 mM and 1 mM) (Figure 1d). Intriguingly, in the PANC-1 cell line, cells treated with Can 100 μM, Dap 100 μM, and Met (5 μM, 50 μM and 500 μM) showed a significant increase in proliferation rates, while in cells treated with Cis (0.5 and 1 mM), a significant decrease in proliferation was observed (Figure 1e). As for the HDFa cell line, cells treated with Dap (1 μM, 10 μM and 100 μM), Can (100 μM), and Cis (1 mM) showed a significant concentration-dependent decrease in proliferation rates (Figure 1f).

### 2.2. Effects of Canagliflozin (Can), Dapagliflozin (Dap), or Metformin (Met) on ^3^H-DG Cellular Uptake, Cell Migration, Cell Cycle, and Cell Apoptosis

Based on the previous results, we decided to choose a concentration of each drug to carry out the next experiments in the two cancer cell lines. The concentrations chosen were 10, 100, and 500 µM for Can, Dap, and Met, respectively. These were the concentrations selected because, of all the concentrations tested, they were the ones that showed cytotoxic effects in the tumor cell lines but not in the normal cell line. We also included the reference compound, Cis.

The effect of the treatments on ^3^H-DG uptake was evaluated. In the AsPC-1 cell line treated with Met 500 μM, a significant increase in ^3^H-DG uptake was observed (Figure 2a). In contrast, in the PANC-1 cell line, there was a marked decrease in ^3^H-DG uptake in cells treated with Cis 1 mM (Figure 2b).

Figure 2c,d show the impact of the treatments on migration ability. In the AsPC-1 cell line (Figure 2c and Appendix A), there was a significant decrease in cell migration in cells treated with Cis 1 mM. The same was observed for the PANC-1 cell line (Figure 2d and Appendix A).

In relation to the effect of Can, Dap, and Met on the cell cycle of the two pancreatic cancer cell lines, no significant effects were observed in the AsPC-1 cell line (Figure 3a and Appendix A). However, in the case of the PANC-1 cell line, a significant decrease in the percentage of cells in the S phase, when treated with Can 10 μM, was found (Figure 3b and Appendix A).

The effect of treatments on the apoptosis rates of the two cell lines was also assessed. In relation to the AsPC-1 cell line, cells treated with Dap 100 μM and Met 500 μM showed a significant decrease in the number of cells in necrosis (quadrant Q1). In addition, there was a significant reduction in the number of viable cells (quadrant Q4) for cells treated with Can 10 μM (Figure 3c and Appendix A). In the PANC-1 cell line, treatment with Dap 100 μM or Met 500 μM caused a significant increase in the percentage of cells in early apoptosis (quadrant Q3) (Figure 3d and Appendix A).

### 2.3. Influence of Inhibitors of Intracellular Signaling Pathways on Cytotoxic Effects of Canagliflozin (Can), Dapagliflozin (Dap), or Metformin (Met)

We next investigated if specific intracellular signaling mechanisms are involved in the cytotoxic effect of Can, Dap, and Met in AsPC-1 and PANC-1 cells. For this, we evaluated the effect of inhibitors of five distinct intracellular signaling pathways (ERK, JNK, mTOR, PI3K, and p38MAPK) in altering cell viability in combination with Can, Dap, and Met.

Starting with the AsPC-1 cell line, we verified that the ERK inhibitor potentiates the cytotoxic effect of Can, Dap, and Met (Figure 4a). As for the JNK inhibitor (Figure 4b) and the PI3K inhibitor (Figure 4c), both are able to abolish the cytotoxic effect of Can. In addition, the JNK inhibitor is also able to abolish the effect of Dap. As for the mTOR inhibitor, its combination with Can, Dap, or Met does not cause any significant change in their cytotoxicities (Figure 4d). Finally, the p38 MAPK inhibitor potentiates the cytotoxic effect of Dap and Met (Figure 4e).

Of note, none of the inhibitors alone affected the viability of the AsPC-1 cell line (Figure 4).

In relation to the PANC-1 cell line, the ERK inhibitor (Figure 5a) and the p38 MAPK inhibitor (Figure 5e) do not cause any change in the cytotoxicities of Can, Dap, or Met. On the other hand, the mTOR inhibitor (Figure 5c), and the PI3K inhibitor (Figure 5d), when combined with Can, Dap, or Met, significantly increased their cytotoxic effects, and the JNK inhibitor potentiated the cytotoxic effect of Can (Figure 5b).

Of note, apart from the PI3K inhibitor, none of the other inhibitors alone affected the viability of the PANC-1 cell line (Figure 5).

### 2.4. Effects of Canagliflozin (Can) or Dapagliflozin (Dap) in Combination with Metformin (Met) on Cell Viability, ^3^H-DG Cellular Uptake, Cell Migration, Cell Cycle, and Cell Apoptosis

Subsequently, to see if there was any addition or synergy between the different treatments, Can 10 μM and Dap 100 μM were each combined with Met 500 μM, and the impact of these combinations on cell viability, ^3^H-DG uptake, cell migration, cell cycle distribution, and apoptosis was assessed in the two cell lines, AsPC-1 and PANC-1.

Figure 6a,b show that, in both cell lines, the combinations Can 10 μΜ + Met 500 μΜ and Dap 100 μΜ + Met 500 μΜ significantly reduced cell viability compared to the control. Furthermore, in the AsPC-1 cell line (Figure 6a), the combination Can 10 μΜ + Met 500 μΜ further reduces cell viability when compared to Can 10 μΜ alone.

In relation to ^3^H-DG uptake, the combination Can 10 μΜ + Met 500 μΜ has no significant effect on ^3^H-DG uptake, in both cell lines (Figure 6c,d). As for Dap 100 μΜ + Met 500 μΜ, this combination significantly increases ^3^H-DG uptake in the AsPC-1 cell line (Figure 6c).

In relation to cell migration, there was no effect of the treatments, separately or in combination, in the AsPC-1 cell line (Figure 6e and Appendix A). Interestingly, in the PANC-1 cell line, a significant decrease in cell migration was observed in the Dap 100 μM + Met 500 μM conditions, with no effects of the treatments separately (Figure 6f and Appendix A).

As for cell cycle analysis, AsPC-1 cells treated with Dap 100 μM + Met 500 μM had a significantly higher percentage of cells in the G1 phase and a lower percentage of cells in the S phase when compared to the control (Figure 7a and Appendix A). The same effect was observed in the PANC-1 cell line, with the combinations Can 10 μM + Met 500 μM and Dap 100 μM + Met 500 μM (Figure 7b and Appendix A).

Finally, the results of the apoptosis assay showed, in the AsPC-1 cell line, a significant decrease in the number of viable cells (quadrant Q4) in cells treated with Can 10 μM + Met 500 μM, when compared to cells treated with Can 10 μM alone (Figure 7c and Appendix A). Both combinations, in the PANC-1 cell line, increased the number of cells in early apoptosis (quadrant Q3) and decreased the number of live cells (quadrant Q4), both having a greater effect than the compounds alone (Figure 7d and Appendix A).

### 2.5. Influence of Inhibitors of Intracellular Signaling Pathways on the Cytotoxic Effects of Canagliflozin (Can) or Dapagliflozin (Dap) in Combination with Metformin (Met)

Finally, the ability of inhibitors of intracellular signaling pathways (the same inhibitors mentioned in Section 2.3) in altering the effect of the combination of Dap or Can with Met on AsPC-1 and PANC-1 cell viability was investigated. Some of the results have already been presented above in Section 3; so, in this section, we will focus on the conditions Can 10 μM + Met 500 μM and Dap 100 μM + Met 500 μM, separately and in combination, with the inhibitor.

Starting with the AsPC-1 cell line (Figure 8), the presence of the ERK inhibitor potentiated the cytotoxic effect of the two combinations (Figure 8a). On the other hand, inhibitors of JNK (Figure 8b), mTOR (Figure 8c), and PI3K (Figure 8d) did not significantly change the cytotoxicity of the combinations Can 10 μM + Met 500 μM and Dap 100 μM + Met 500 μM. Finally, concerning the p38 MAPK inhibitor, this inhibitor potentiated the cytotoxic effect of the combination Dap 100 μM + Met 500 μM, but it did not influence the effect of the combination Can 10 μM + Met 500 μM (Figure 8e).

Finally, analysis of the results for the PANC-1 cell line showed that ERK (Figure 9a), JNK (Figure 9b), PI3K (Figure 9d), and p38 MAPK (Figure 9e) inhibitors do not significantly change the cytotoxic effect of both combinations. In contrast, the cytotoxic effect of both combinations is potentiated by the mTOR inhibitor (Figure 9c).

## 3. Discussion

PC is one of the leading causes of cancer-related death [1,2]. Currently, the main treatment option remains surgical resection, but this cancer still has high morbidity, mortality, and recurrence rates [24]. This is because PC is often diagnosed at a very advanced stage [8] and because this neoplasm is highly complex due to its molecular diversity and actionable mutations, which makes it difficult to achieve a positive response to treatment [25]. In addition, the tumor microenvironment of PC is immunosuppressive, making it difficult for therapies commonly used in other types of cancer, such as immunotherapy and radiotherapy, to be successful [26]. So, developing new therapies to overcome these difficulties is essential, and the concept of drug repositioning has emerged. This strategy consists of identifying new therapeutic uses for existing drugs already approved for other conditions, thus speeding up the development process and reducing the associated costs [27].

According to the Warburg effect, malignant cells consume more glucose than normal cells [28]. This is because they change glucose metabolism to a less efficient pathway in terms of ATP production per molecule of glucose—aerobic glycolysis (oxidation of glucose to lactate)—but that nevertheless confers these cells several advantages. Namely, it helps to support the biosynthetic requirements of uncontrolled proliferation, it presents an advantage for cell growth in a multicellular environment (acidification of the microenvironment), and it confers direct signaling functions to tumor cells [29]. The increased uptake of glucose by cancer cells depends on the increased activity of glucose membrane transporters. Among glucose membrane transporters, several tumor cells, and particularly PC cells, have an increased expression of the SGLT2 co-transporter, and some studies concluded that SGLT2 inhibitors may have a beneficial effect in PC therapy [16,17]. For these reasons, SGLT2 co-transporter inhibitors have emerged as a possible therapeutic option for this malignant neoplasm. Furthermore, it is well known that individuals with T2DM are more likely to develop PC [30]. In T2DM individuals, Met is used as a first-line therapy, either separately or in combination with other drugs, namely SGLT2 inhibitors, and has also been shown to have an anti-tumoral effect [31]. However, nothing is known concerning the effect of the combination of metformin and SGLT2 inhibitors on PC.

Therefore, based on this information, bearing in mind the concept of drug repositioning and the need to find therapeutic alternatives for PC, two SGLT2 inhibitors, Can and Dap, were tested separately and in combination with Met. For comparison purposes, Cis, a drug with a well-described anti-neoplastic effect, was used as a reference compound.

In the first experiments, three different concentrations of each drug were applied separately to three cell lines (two tumor cell lines and a normal cell line), to select a concentration of each drug for the remaining tests and to assess the selectivity of the treatments for tumor cells. Because cancer cells have a greater capacity for survival and a higher proliferation rate, we began by assessing the effect of the treatments on cell viability and proliferation [32]. In relation to the cell viability results, Can was cytotoxic for the three cell lines, but with distinct potencies. In contrast, Dap reduced cell viability in the AsPC-1 cancer cell line only. As for Met, this compound reduced cell viability in the two cancer cell lines (AsPC-1 and PANC-1) but no cytotoxic effect on the HDFa cell line was observed. In terms of cell proliferation, both Can and Dap reduced cell proliferation in AsPC-1 and HDFa cells, but Can, Dap, and Met increased the proliferation rates of the PANC-1 cell line. The reference compound, Cis, was cytotoxic and antiproliferative for the three cell lines.

After analyzing the results of the above-mentioned tests, we concluded that the tested SGLT2 inhibitors have a cytotoxic and anti-proliferative effect on the two PC cell lines, with a more marked effect on the AsPC-1 cell line. These results are in line with those described in the literature, since SGLT2 inhibitors [33] and metformin [34] have shown cytotoxic and antiproliferative effects in several cancer cell types, including PC cell lines. It could also be concluded that Can 10 μM, Dap 100 μM, and Met 500 μM showed the most interesting results, since they were able to reduce cell viability and cell proliferation of the tumor cell lines but had mostly no effect on the normal cell line. So, these were the concentrations chosen to investigate the impact of these compounds on other tumor cell characteristics, such as glucose uptake, migration ability, cell cycle, and apoptosis rates.

As described above, malignant cells consume much more glucose than normal cells [28]. It is known that glucose is the main source of energy for cancer cells, giving them the ability to satisfy their biosynthetic needs and adapt to various microenvironments [34]. For this reason, it is important when analyzing new anti-tumor therapies to assess their effect on glucose uptake. The results obtained show that SGLT2 inhibitors had no effect on ^3^H-DG uptake in both cell lines, and that Met increased ^3^H-DG uptake in the AsPC-1 cell line. A previous study concluded that SGLT2 transporters are overexpressed in PC cells, actively contributing to the transport of glucose into these cells [16]. It was therefore expected that these inhibitors would influence ^3^H-DG uptake, which was not the case. In addition to SGLT2, PC cells also overexpress transporters from the GLUT family [35]. So, one possible explanation for the lack of effect of SGLT2 inhibitors on ^3^H-DG uptake may be the fact that uptake of this compound into these cells is mediated by GLUT. So, it could be important to use an SGLT2-specific glucose analog to check whether glucose uptake is actually SGLT2-mediated, as performed in the previous work [16]. Concerning Met, the results obtained with the PANC-1 cell line are in line with those expected, since a previous study found that this agent did not produce any change in ^18^F-fluorodeoxyglucose uptake in PC cells [36]. However, in the AsPC-1 cell line, Met caused an increase in ^3^H-DG uptake, which may be related to the fact that this is a more aggressive tumor line. Of note, the effect of Met on ^18^F-fluorodeoxyglucose uptake by cancer cell lines, which is exclusively SGLT2-mediated, is variable: it causes either an increase (hepatocellular carcinoma and breast cancer), a decrease (thyroid cancer), or no change (colon and PC) [36]. Also, a study carried out by Amaral et al. on breast tumor cell lines showed that prolonged exposure to Met increased ^3^H-DG uptake by these cells [37].

Another adaptive characteristic developed by cancer cells is their high mobility, which allows them to migrate to another location and metastasize [32]. Separately, none of the treatments produced any significant effect, although a tendency for a decrease in cell migration ability was seen. A previous study found a decrease in the migratory capacity of PANC-1 cells treated with IC_50_ and sub-IC_50_ concentrations of Can [38]. The reason for the discrepant effects is not apparent, given that in the aforementioned study, the IC_50_ value for Can (24 h) in this cell line was 65 μM, and we used 10 µM Can in our experiments. The same discrepancy between our results and previous ones is observed for Met, as Met inhibits the migratory and invasive ability of several tumor cells, including PC cell lines [39].

Another hallmark of cancer cells is their high replication rate, and so it is important to analyze the impact of drugs on the cell cycle [32]. A decrease in the percentage of cells in the S phase in PANC-1 cells treated with Can, and, in both cell lines, a tendency for cell cycle arrest in the G1 phase after treatment with Can, Dap, and Met was found. These results are in agreement with a previous study, which found that Can interrupted the cell cycle in the sub-G1 phase in another pancreas ductal adenocarcinoma cell line (MIA-PaCa-2) [40]. Yamana et al. also found that Met arrested the cell cycle in the G0/G1 phase in a pancreatic neuroendocrine tumor cell line (QGP-1 cells) [41].

Another characteristic that is important to explore when applying new anti-tumor therapies is whether the treatments are increasing the number of cells undergoing apoptosis [32]. In the AsPC-1 cell line, Can, Dap, and Met tended to increase the number of cells in early apoptosis, and, in the PANC-1 cell line, Dap and Met significantly increased the number of cells in early apoptosis (and a similar tendency was observed for Can). The results obtained are in line with previous works, which have shown that Met is capable of inducing apoptosis in PC cells [42,43] and that MIA-PaCa-2 cells died by apoptosis when treated with Can [40].

The results obtained with Can, Dap, and Met on ^3^H-DG uptake point to the conclusion that the anticancer effect of these compounds on the PC cell lines tested does not seem to be related to the inhibition of cellular glucose uptake, but rather to another effect.

While the primary mode of anti-diabetic action of SGLT2 inhibitors has been attributed to suppression of glucose uptake, other mechanisms appear to be involved in their anticancer effect, including interference with intracellular signaling pathways such as the PI3K-Akt pathway [33,44]. Similarly, a very important mechanism responsible for the anticancer effect of Met involves interference with intracellular signaling pathways such as AMP-activated protein kinase (AMPK), PI3K-Akt-mTOR, and mitogen-activated protein kinase (MAPK) [18,45]. We therefore investigated if interference with intracellular signaling pathways could be involved in their cytotoxic effect. We focused on intracellular signaling mechanisms affected by SGLT2 inhibitors (PI3K-Akt, mTOR, and ERK) [32,43] and Met (PI3K-Akt, mTOR, ERK, JNK, and p38 MAP kinase) [18,45].

The results obtained suggest that the pathways by which the tested SGLT2 inhibitors (but not Met) induce their toxicity in AsPC-1 cells are the JNK pathway and the PI3K pathways. The JNK pathway is one of the main MAPK signaling pathways and is involved in the cellular response to various stress stimuli and in regulating cellular processes such as apoptosis, cell differentiation, proliferation, and inflammation [46]. The PI3K pathway in turn plays a crucial role in regulating various cellular processes such as growth, proliferation, survival, metabolism, and cell motility [47]. In agreement with our results, two recent works found that Can affects the PI3K pathway in hepatic [48] and PC cell lines [49].

Additionally, our results also showed an additive effect of ERK and p38MAPK inhibitors and the SGLT2 inhibitors or Met in the AsPC-1 cell line, and an additive effect of JNK, PI3K, and mTOR inhibitors and the SGLT2 inhibitors or Met in the PANC-1 cell line. This additive relationship suggests distinct mechanisms of cytotoxicity of these compounds. In this context, the fact that the PI3K inhibitor alone produced a cytotoxic effect on the PANC-1 cell line shows that this is an intracellular pathway crucial for survival of these cells.

In the final part of this work, we assessed whether there was any additive anticancer effect when combining SGLT2 inhibitors and Met, by analyzing the impact of their combination on some of the carcinogenic properties already assessed.

We verified that the Can+Met combination was more cytotoxic than Can alone in the AsPC-1 cell line, and that the Dap+Met combination caused a significant reduction in the migratory ability of PANC-1 cells, which was not observed with either of these compounds alone. So, we demonstrate the existence of some additive cytotoxic and antimigratory effects between these two compounds in PC cell lines.

As far as ^3^H-DG uptake is concerned, similarly to the compounds alone, the combinations had no inhibitory effect on this parameter. As mentioned before, the reason for this observation might be related to the overexpression of transporters from the GLUT family, which is an alternative pathway for glucose uptake, in these cells [16,35].

In the cell cycle assay, the combination of Dap+Met increased the number of cells in the G1 phase and decreased the number of cells in the S phase, for both cell lines. A similar result was observed with the Can+Met combination in the PANC-1 cell line. Because separately the compounds had no significant effect on the cell cycle, we conclude that, in both cell lines, there is an addition between the effect of treatments resulting in G1 phase arrest.

Finally, in relation to apoptosis, the combinations Can+Met and Dap+Met increased the number of cells undergoing early apoptosis in the PANC-1 cell line. Because this was not observed with the compounds separately, an additive effect appears to exist.

Lastly, we checked the effect of the inhibitors of intracellular signaling pathways on the cytotoxic effect caused by the combinations of SGLT2 inhibitors and Met. Unlike what was observed with the isolated compounds, none of the inhibitors were able to abolish the cytotoxic effect of either the combination of Can+Met or Dap+Met, in both cell lines. Furthermore, their cytotoxic effect was even more marked when associated with inhibition of ERK (Can+Met and Dap+Met) and p38MAPK (Dap+Met) in AsPC-1 cells, and when associated with inhibition of mTOR (Can+Met and Dap+Met) in PANC-1 cells. These results led us to conclude that the SGLT2 inhibitors and Met have different mechanisms of anticancer action, because, even if the effect of one of the compounds is inhibited, the other compound will continue to act, producing the cytotoxic effect. However, the fact that in some cases the cytotoxic effect is potentiated by the combination with the inhibitor may suggest a possible additive effect between them.

In conclusion, SGLT2 inhibitors were found to be cytotoxic, antiproliferative, and pro-apoptotic in the PC cell lines. In general, the cytotoxic effect, the antimigratory effect, the effect on the cell cycle (G1 phase arrest), and the pro-apoptotic effect were enhanced when the SGLT2 inhibitor was combined with Met. Moreover, the cytotoxic effects of Dap and Can in AsPC-1 cells involve PI3K- and JNK-dependent pathways. In contrast, the cytotoxic effects of Can+Met and Dap+ Met were not sensitive to PI3K or JNK inhibition. Therefore, in the future, more tests are needed in order to understand how SGLT2 inhibitors (and Met) produce their antitumoral effects in PC.

## 4. Materials and Methods

### 4.1. Cell Culture

Three cell lines were used in this study: one normal cell line, the HDFa human dermal fibroblast cell line (ATCC PCS-201-012, passage numbers 19–32), and two PC cell lines, the AsPC-1 (ATCC CRL-1682; passage numbers 9–33) and the PANC-1 (ATCC CRL-1469™, passage numbers 12–48) cell lines. The AsPC-1 cell line is a PC cell line with high metastatic potential, initiated from cells derived from ascites of a 62-year-old woman with pancreatic cancer [50,51]. The PANC-1 cell line is a PC cell line with a low metastatic rate, isolated from the pancreatic duct of a 56-year-old man with an adenocarcinoma [52].

The HDFa and PANC-1 cell lines were maintained in Dulbecco’s modified Eagle’s medium (DMEM, PAN-BIOTECH™—P04-045110, Aidenbach, Germany) with 0.3 g/L L-glutamine; 4.5 g/L glucose; 3.7 g/L NaHCO_3_ supplemented with 10% fetal bovine serum (FBS, PAN-Biotech™); and 1% penicillin/streptomycin/amphotericin B solution (PAN-Biotech™). The AsPC-1 cell line was maintained in Roswell Park Memorial Institute medium (RPMI, Sigma R6504, Burlington, MA, USA) with 0.3 g/L L-glutamine, 2 g/L D-glucose supplemented with 1 mM sodium pyruvate; 10 mM NaHCO_3_, 10 mM HEPES; 10% FBS (PAN-Biotech™); and 1% penicillin/streptomycin/amphotericin B solution (PAN-Biotech™). The cell lines were maintained in an incubator (Thermo Fisher Scientific, Waltham, MA, USA, EUA) at 37 °C, with a humid atmosphere and 5% CO_2_.

The cells were cultured in culture plates (21 cm^2^; ∅ 60 mm; Corning Costar, Corning, NY, USA), and the culture medium was changed every two to three days. When the culture reached 90% confluence, a split protocol was carried out (1:3 dilution).

### 4.2. Cell Treatments

The cells were treated for 24 h with different concentrations of the drugs, added to FBS-free culture medium: Can (Cayman Chemicals, Ann Arbor, MI, USA); Dap (Cayman Chemicals); Met; and Cis (Merck, Darmstadt, Germany). The two selective SGLT2 inhibitors, Can and Dap, were tested at 1 μM, 10 μM, and 100 μM and were dissolved in DMSO. Met was tested at concentrations of 5 μM, 50 μM, and 500 μM and was dissolved in water. Finally, Cis was tested at concentrations of 0.5 mM and 1 mM and was dissolved in DMSO. The drugs were first tested separately and then in combination (10 μM Can combined with 500 μM Met, and 100 μM Dap combined with 500 μM Met). The final concentration of their respective solvents in the culture medium was 1% (*v*/*v*).

Controls were run in the presence of the solvents only. In some experiments, the cells were also treated with intracellular signaling pathway inhibitors. The inhibitors used and their concentrations were as follows: extracellular-signal-regulated kinase inhibitor (ERKi; PD 98059 2.5 µM), p38 mitogen-activated protein kinase inhibitor (p38 MAPKi; SB 203580 2.5 µM), c-Jun N-terminal kinase inhibitor (JNKi; SP 600125 5 µM), phosphoinositide 3-kinase inhibitor (PI3Ki; LY 294002 1 µM), and mammalian target of rapamycin inhibitor (mTORi; rapamycin 100 nM) (all from Sigma). The final concentration of the respective solvents in the culture medium was 1% (*v*/*v*). Controls were carried out only in the presence of the solvents.

### 4.3. Evaluation of the Cell Viability—MTT Assay

Cell viability was assessed using the 3-(4,5-dimethylthiazol-2-yl)-2,5-diphenyl tetrazolium bromide (MTT) test. To carry out the test, the cells were seeded in a 96-well plate (TPP^®^, Trasadingen, Switzerland) and left to incubate in a humidified incubator at 37 °C until they reached around 70–80% confluence. After reaching the required confluence, the culture medium was removed and 100 μL of FBS-free culture medium with treatment was added. After 24 h, 10 μL of MTT reagent (Sigma) was added to each well and left to act for 1 h. After 1 h, the medium present in all the wells was aspirated and 100 μL of DMSO was added to dissolve the crystals formed. The absorbance was then measured at 550 nm and 650 nm on a microplate reader (Thermo Fisher Scientific). Viability was determined by the difference between abs 550 nm and abs 650 nm and expressed as a % of the control.

### 4.4. Evaluation of Cell Proliferation Rates—Incorporation of ^3^H-Thymidine Assay

Cell proliferation was assessed by the incorporation of ^3^H-thymidine ([methyl-^3^H]-thymidine; specific activity 69.2 Ci/mmol) (GE Healthcare GmbH, Freiburg, Germany). To carry out the test, the cells were seeded in a 24-well plate (2 cm^2^, TPP^®^) and left to incubate in a humidified incubator at 37 °C until they reached around 70–80% confluence. Once they had reached the required confluence, the culture medium was removed and 200 μL of FBS-free culture medium with treatment was added to each well. After 19 h, the applied treatment was removed and 200 μL of FBS-free culture medium containing ^3^H-thymidine (0.525 µCi/mL) and the respective treatments were added to each well, reserving one well of each treatment for protein quantification, to which 300 μL Triton X-100 (Merck) was added. After 5 h of incubation with ^3^H-thymidine, the wells containing Triton X-100 were scraped and the entire volume was placed in Eppendorfs for subsequent protein quantification using the Bradford method [53], by reading the absorbance at 595 nm. The remaining wells containing incubation medium and the respective treatments were aspirated and then washed with 300 μL of 10% trichloroacetic acid (TCA). After washing, the plate was dried at room temperature and 280 μL of 1 M NaOH was added to each well for 15 min. Finally, 250 μL of each well was pipetted into vials previously prepared with 50 μL of 5 M HCL and 2 mL of scintillation mixture, taking care to prepare the blank by adding only 100 μL of incubation medium. Radioactivity was measured by liquid scintillometry (LKB Wallac Liquid Scintillation Counter 1209, Turku, Finland), and the rate of DNA synthesis was expressed as ^3^H-thymidine incorporation (µCi/mg protein).

### 4.5. Evaluation of Cell Migration—Wound Healing Assay

Cell migration was assessed using the wound healing (scratch injury) assay. To carry out the test, the cells were seeded in a 24-well plate (2 cm^2^, TPP^®^) and left to incubate in a humidified incubator at 37 °C until they reached 100% confluence. After reaching this state of confluence, a wound was made in the cell monolayer using a 10 μL micropipette tip, and 300 μL of FBS-free culture medium with treatment was added. Images were taken at 0 h and 24h after scratching (Motic AE31E microscope, Xiamen, China), and the injured area was quantified using ImageJ software version 2.14.0 (NIH, Bethesda, MD, USA). Results are presented as % of migration (% of total area).

### 4.6. Evaluation of Cellular ^3^H-Deoxy-D-Glucose (^3^H-DG) Uptake

The effect of the compounds on the cellular uptake of glucose was assessed by quantifying the uptake of a glucose analog, ^3^H-deoxy-D-glucose (2-[1,2-^3^H(N)]-deoxy-D-glucose: specific activity 60 Ci/mmol) (^3^H-DG) (American Radiolabeled Chemicals Inc., St Louis, MO, USA). To carry out the test, the cells were seeded in a 24-well plate (2 cm^2^, TPP^®^) and left to incubate in a humidified incubator at 37 °C until they reached 100% confluence. After reaching the required state of confluence, the culture medium was removed and 300 μL of FBS-free culture medium with treatment was added to each well for 24 h. After 24 h, the plates were placed in a 37 °C water bath, and the cells were washed with 300 µL of GF-HBS buffer (composition in mM: 20 HEPES, 5 KCl, 140 NaCl, 2.5 MgCl_2_, 1 CaCl_2_, pH 7.4). After washing, a pre-incubation was carried out at 37 °C in GF-HBS buffer with the respective treatments for 20 min, followed by incubation with 200 µL of ^3^H-DG 10 nM in GF-HBS buffer with the respective treatments for 6 min. Finally, the reaction was stopped by rinsing the cells with 500 µL of ice-cold GF-HBS buffer and adding 300 µL of Triton X-100 (0.1%). Subsequently, 250 µL of each sample was transferred to a vial with 2 mL of scintillation liquid, and the radioactivity was measured by liquid scintillation counting (LKB Wallac 1209 Rackbeta). The results were normalized to the total protein content.

### 4.7. Cell Cycle Evaluation

The effect of the compounds on the cell cycle was assessed by flow cytometry. To carry out the tests, the cells were seeded in 21 cm^2^ plates (TPP^®^) and left to incubate in a humidified incubator at 37 °C until they reached 80% to 90% confluence. After reaching the required state of confluence, the culture medium was removed and 3 mL of FBS-free culture medium with treatment was added to each well, leaving it to act for 24 h. After 24 h, a split protocol was carried out and 1 mL of the cell suspension was transferred to an Eppendorf. The cells were then centrifuged for 5 min at 300× *g* and fixed in 200 μL of 70% ethanol for 30 min at 4 °C. After this step, 1 mL of PBS with 2% bovine serum albumin (BSA) was added to each Eppendorf and centrifuged again for 5 min at 300× *g*. Finally, 85 µL of propidium iodide (PI) solution (PI/Rnase, Immunostep, Salamanca, Spain) was added for 15 min at room temperature. Flow cytometry analysis was performed on a BD Accuri C6 PLUS flow cytometer (Becton, Dickinson and Company, Franklin Lakes, NJ, USA). The results were analyzed using FlowJo software version 10.10.0 (Ashland, OR, USA). Results are expressed as % of total cells arrested in each phase in the cell cycle.

### 4.8. Apoptosis Analysis

The effect of the compounds on the apoptosis was assessed by flow cytometry. To carry out the tests, the cells were seeded in 21 cm^2^ plates (TPP^®^, Trasadingen, Switzerland) and left to incubate in a humidified incubator at 37 °C until they reached 70% to 80% confluence. After reaching the required state of confluence, the culture medium was removed and 3 mL of FBS-free culture medium with treatment was added to each well, leaving it to act for 24 h. After 24 h, the apoptotic cells were quantified using apoptosis detection kit FITC (88-8005-74; ThermoFisher), according to the manufacturer’s instructions. Briefly, cells were collected and washed with PBS and then, once, in 1× binding buffer. Next, cells were resuspended in 1× binding buffer containing 5 µL annexin V-FITC for 15 min in the dark. Lastly, the cells were washed once in 1× binding buffer and resuspended in 400 μL 1× binding buffer containing 5 μL PI. The samples were immediately analyzed using a BD Accuri C6 PLUS flow cytometer (Becton, Dickinson and Company, Franklin Lakes, NJ, USA). The results were analyzed using FlowJo software version 10.10.0 (Ashland, OR, USA).

### 4.9. Statistics

The results are presented as arithmetic means with the standard error of the mean (SEM). The n indicates the number of replicates in the total of the trials. Statistical significance between three or more groups was assessed by a one-way ANOVA test, followed by the Tukey test. Student’s *t*-test was used to compare the two groups. Differences were considered significant when *p* < 0.05. Analyses were carried out using GraphPad Prism software version 8.0 (San Diego, CA, USA).

## Figures and Tables

**Figure 1 ijms-25-09932-f001:**
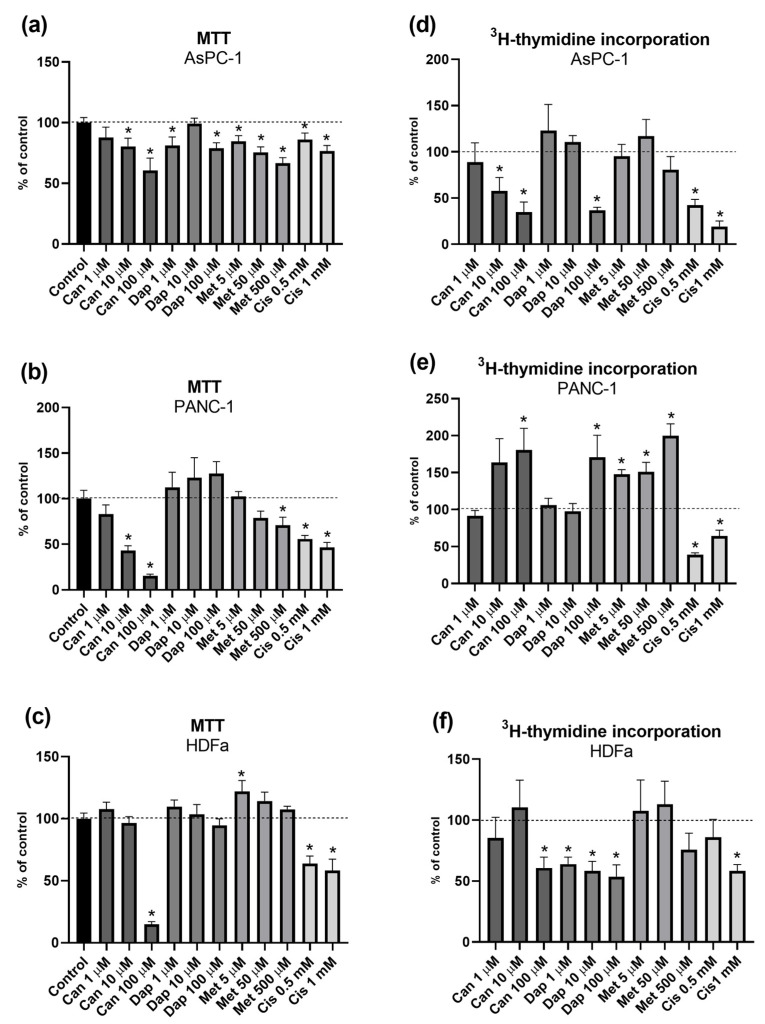
Effect of treatments with canagliflozin (Can), dapagliflozin (Dap), metformin (Met), and cisplatin (Cis) on cellular viability of (**a**) AsPC-1 (n = 19; 4 trials), (**b**) PANC-1 (n = 12; 3 trials), and (**c**) HDFa (n = 19; 4 trials) cell lines, and on proliferation rates of (**d**) AsPC-1 (n = 9; 3 trials), (**e**) PANC-1 (n = 9; 3 trials), and (**f**) HDFa (n = 12; 4 trials) cell lines. Results are expressed as arithmetic means ± SEM. * Significantly different from control (*p* < 0.05).

**Figure 2 ijms-25-09932-f002:**
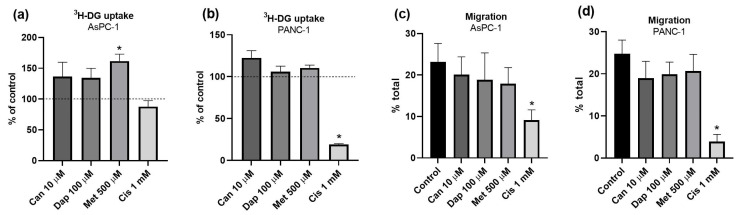
Effect of treatments with canagliflozin (Can), dapagliflozin (Dap), metformin (Met), and cisplatin (Cis) on ^3^H-DG uptake by (**a**) AsPC-1 (n = 8; 2 trials) and (**b**) PANC-1 (n = 8; 2 trials) cell lines, and on migration ability of (**c**) AsPC-1 (n = 18; 3 trials) and (**d**) PANC-1 (n = 18; 3 trials) cell lines. Results are expressed as arithmetic means ± SEM. * Significantly different from the control (*p* < 0.05).

**Figure 3 ijms-25-09932-f003:**
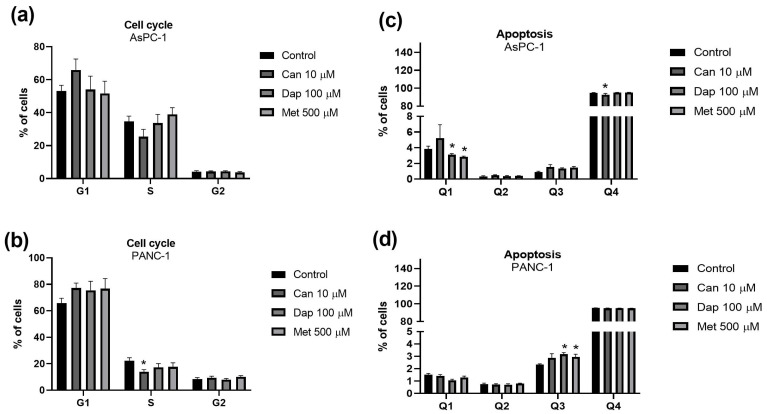
Effect of treatments with canagliflozin (Can), dapagliflozin (Dap), and metformin (Met) on the cell cycle in the (**a**) AsPC-1 (n = 6; 2 trials) and (**b**) PANC-1 (n = 6; 2 trials) cell lines and on apoptosis rates in the (**c**) AsPC-1 (n = 6; 2 trials) and (**d**) PANC-1 (n = 6; 2 trials) cell lines. Results are expressed as arithmetic means ± SEM. * Significantly different from the control (*p* < 0.05).

**Figure 4 ijms-25-09932-f004:**
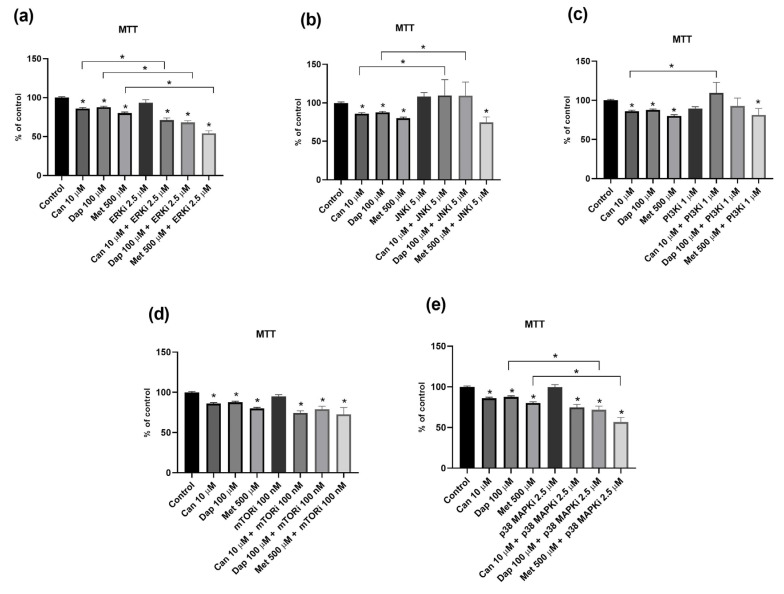
Influence of intracellular signaling pathway inhibitors on the cytotoxic effect of canagliflozin (Can), dapagliflozin (Dap), and metformin (Met) on the AsPC-1 cell line. (**a**) Inhibitor of extracellular-signal-regulated kinases (ERKi; PD 98059) (n = 8; 2 trials); (**b**) inhibitor of c-Jun N-terminal kinase (JNKi; SP 600125) (n = 8; 2 trials); (**c**) inhibitor of phosphoinositide 3-kinase (PI3Ki; LY 294002) (n = 12; 3 trials); (**d**) inhibitor of mammalian target of rapamycin (mTORi; rapamycin) (n = 12; 3 trials); and (**e**) inhibitor of p38 mitogen-activated protein kinases (p38 MAPKi; SB 203580) (n = 8; 2 trials). Results are expressed as arithmetic means ± SEM. * Significantly different (*p* < 0.05).

**Figure 5 ijms-25-09932-f005:**
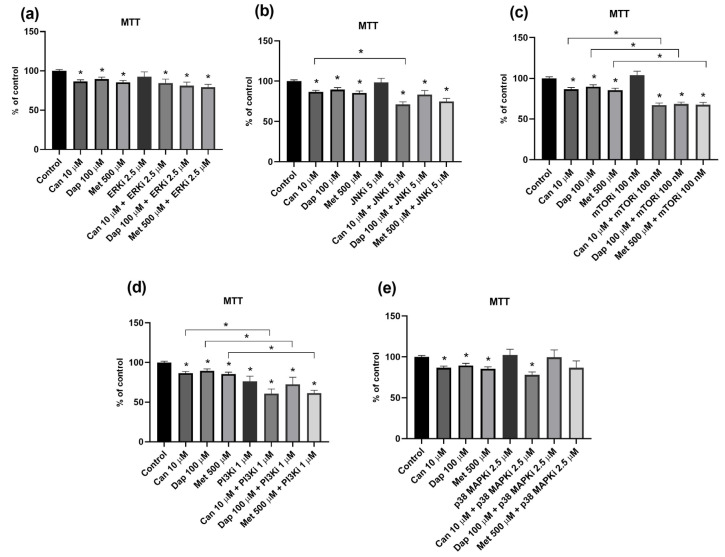
Influence of intracellular signaling pathway inhibitors on the cytotoxic effect of canagliflozin (Can), dapagliflozin (Dap), and metformin (Met) on the PANC-1 cell line. (**a**) Inhibitor of extracellular-signal-regulated kinases (ERKi; PD 98059) (n = 12; 3 trials); (**b**) inhibitor of c-Jun N-terminal kinase (JNKi; SP 600125) (n = 12; 3 trials); (**c**) inhibitor of mammalian target of rapamycin (mTORi; rapamycin) (n = 12; 3 trials); (**d**) inhibitor of phosphoinositide 3-kinase (PI3Ki; LY 294002) (n = 12; 3 trials); and (**e**) inhibitor of p38 mitogen-activated protein kinases (p38 MAPKi; SB 203580) (n = 12; 3 trials). Results are expressed as arithmetic means ± SEM. * Significantly different (*p* < 0.05).

**Figure 6 ijms-25-09932-f006:**
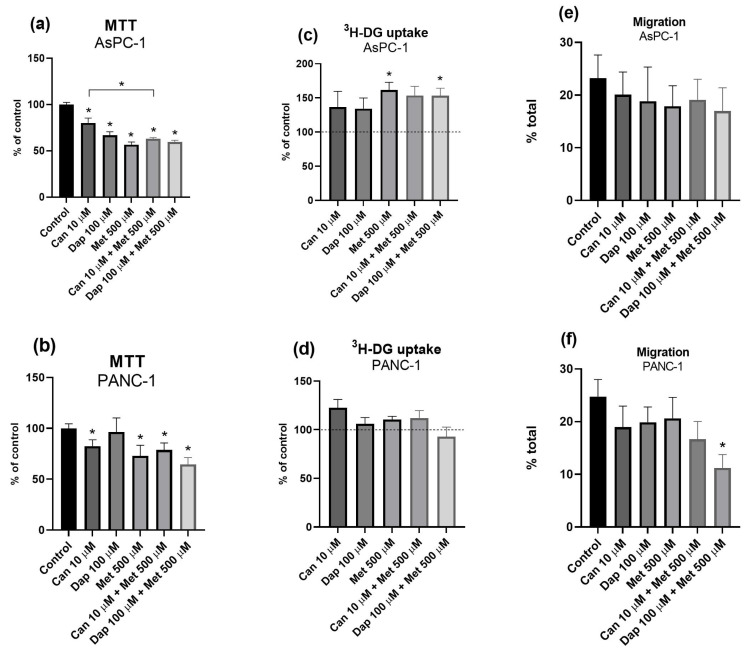
Effects of treatments with canagliflozin (Can), dapagliflozin (Dap), and metformin (Met), separately or in combination, on the viability of the (**a**) AsPC-1 (n = 8; 2 trials) and (**b**) PANC-1 (n = 8; 2 trials) cell lines; on ^3^H-DG uptake by the (**c**) AsPC-1 (n = 8; 2 trials) and (**d**) PANC-1 (n = 8; 2 trials) cell lines; and on migration of the (**e**) AsPC-1 (n = 18; 3 trials) and (**f**) PANC-1 (n = 18; 3 trials) cell lines. Results are expressed as arithmetic means ± SEM. * Significantly different (*p* < 0.05).

**Figure 7 ijms-25-09932-f007:**
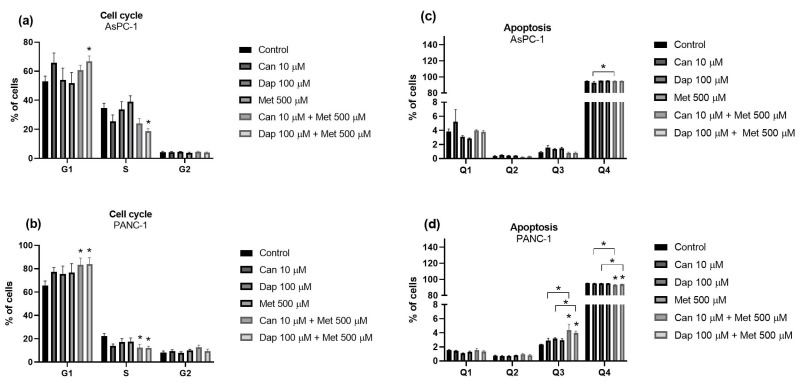
Effects of treatments with canagliflozin (Can), dapagliflozin (Dap) and metformin (Met), separately or in combination, on the cell cycle of the (**a**) AsPC-1 (n = 6; 2 trials) and (**b**) PANC-1 (n = 6; 2 trials) cell lines and on apoptosis of the (**c**) AsPC-1 (n = 6; 2 trials) and (**d**) PANC-1 (n = 6; 2 trials) cell lines. Results are expressed as arithmetic means ± SEM. * Significantly different (*p* < 0.05).

**Figure 8 ijms-25-09932-f008:**
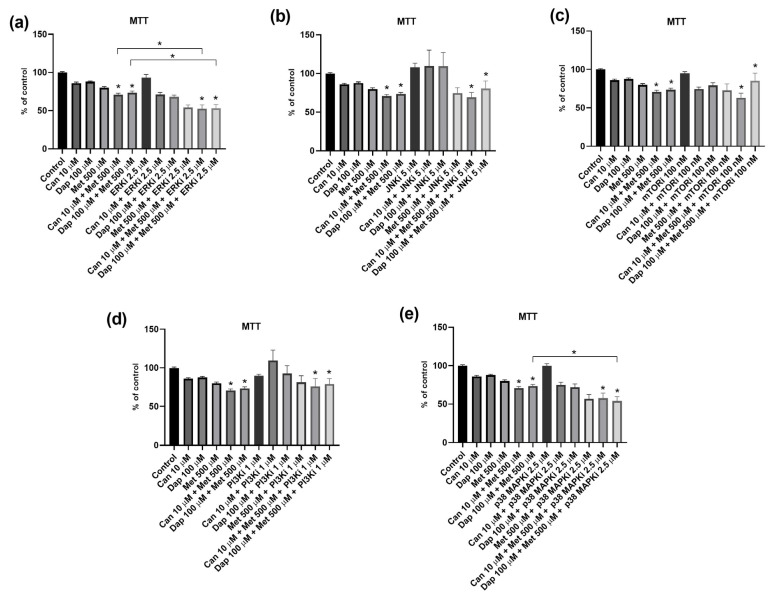
Influence of intracellular signaling pathway inhibitors on the cytotoxic effect of canagliflozin (Can), dapagliflozin (Dap), and metformin (Met), in combination, on the AsPC-1 cell line. (**a**) Inhibitor of extracellular-signal-regulated kinases (ERKi; PD 98059) (n = 8; 2 trials); (**b**) inhibitor of c-Jun N-terminal kinase (JNKi; SP 600125) (n = 8; 2 trials); (**c**) inhibitor of mammalian target of rapamycin (mTORi; rapamycin) (n = 12; 3 trials); (**d**) inhibitor of phosphoinositide 3-kinase (PI3Ki; LY 294002) (n = 12; 3 trials); and (**e**) inhibitor of p38 mitogen-activated protein kinases (p38 MAPKi; SB 203580) (n = 8; 2 trials). Results are expressed as arithmetic means ± SEM. * Significantly different (*p* < 0.05).

**Figure 9 ijms-25-09932-f009:**
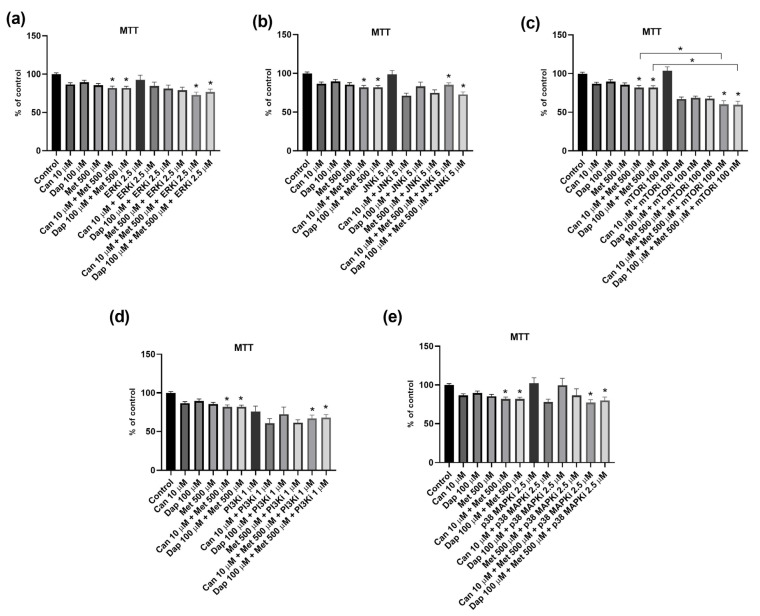
Influence of intracellular signaling pathway inhibitors on the cytotoxic effect of canagliflozin (Can), dapagliflozin (Dap), and metformin (Met) on the PANC-1 cell line. (**a**) Inhibitor of extracellular-signal-regulated kinases (ERKi; PD 98059) (n = 12; 3 trials); (**b**) inhibitor of c-Jun N-terminal kinase (JNKi; SP 600125) (n = 12; 3 trials); (**c**) inhibitor of mammalian target of rapamycin (mTORi; rapamycin) (n = 12; 3 trials); (**d**) inhibitor of phosphoinositide 3-kinase (PI3Ki; LY 294002) (n = 12; 3 trials); and (**e**) inhibitor of p38 mitogen-activated protein kinases (p38 MAPKi; SB 203580). Results are expressed as arithmetic means ± SEM. * Significantly different (*p* < 0.05).

## Data Availability

The data presented in this study are available on request from the corresponding author.

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
