# Peer review of "Effect of Sodium-Glucose Co-Transporter 2 Inhibitors Combined with Metformin on Pancreatic Cancer Cell Lines"

_ijms, 2024, doi:10.3390/ijms25189932_

Round 1

Reviewer 1 Report

Comments and Suggestions for Authors

1.     In figure 1, the significance difference indicates that the comparison relationship is not so clear and cannot be visually seen through the graph, it can be represented by bracket totally.

2.     In figure 3g and h, the clarity of the apoptosis diagram is insufficient, precluding the visualization of the apoptosis ratios within each quadrant.

3.     Significant differences indicate inconsistencies, it should be uniformly expressed as “P0.05” or “P0.5”.

Reviewer 2 Report

Comments and Suggestions for Authors

Review report to ijms-3152231 (Effect of SGLT2 inhibitors combined with metformin on pan-creatic cancer cell lines. André Cristovão, Nelson Andrade, Fátima Martel and Cláudia Silva)

General description: The authors investigated the effects of SGLT2 inhibitors alone and in combination with metformin on PC cell lines. The manuscript is 18 pages without references included 7 figures. The manuscript is in IJMS template.

General suggestions:

1. Difficult to follow the text due to the many small details. Should simplify where it possible – see my suggestion for change in the abstract. Similarly difficult to follow the results and the discussion as well. Also, the discussion should be different then a repetition of the results with some comparison with other manuscript. In the discussion, you should not use as much detail as in the result, which is good there but not here. Should highlight and take the essence of your results; what is new or more in your work compared to others. Also, if there are differences compared to others, have to evaluate why you did not use these information when you planned your experiments (i.e concentration of drugs), or what could be the reason the different results (in some part were presented well).

1. Your new finding is related to the combination treatment. Thus, highlight those both in the abstract and in the main text. Therefore, do not focus too long on single treatments – add supplementary files for the flow pictures and put only in one picture all the single treatment results for selection.

2. Use supplementary files for the presentation of the non-significant changes.

3. Figures are too small, need changes – see below in “format change suggestion”.

4. Figure legends are too busy (drug names, concentration, subpart labelling etc.). If the figure enlarge enough and the titles of the subpart and the concentrations are there, no need to write down the concentrations into the figure legends (only drug names, test name and cell line enough).

5. If I know well 6 figures/tables (all together) are accepted. Please check if allowed to be more then 6. (Use Supplementary files – need rearranging of the results though then.)

Suggestions for the format change

Supplementary file suggested for non-significant, or not so important parts (i.e. flow pictures in cell cycle). However, if the content would stay please see suggestions for changes:

1. Figure 1: the labellings and the signs of the significance are too small. Could be increased the size of the subparts if the orientation would be 3x2 rather than 2x3. If the a,b,c/d,e,f would be vertically arranged then only 2 subfigures/page with will be presented and can enlarge the subfigures with labelling etc– thus the same cell lines could go side by side as well.

2. Figure 2: please, place the “H-DG uptake” and “Migration” as title of the a/b and c/d, respectively, and on the vertical axis just leave the text, which is now in the brackets. More easily to absorb the meaning of the results. Additionally, please label as title of e, and f, which cell lines are presented on which panels. Please also mention this in the figure legend as well.

3. Figure 3: similar changes then were mentioned earlier. Too small column statistics, labelling etc. Similar vertical arranging as was mentioned above could help: a, b, d, e vertically below to each other, enlarge them, and relabel them by a, b, c, d (no need e-h). Then, on the “right side of the picture” add in four square orientation the pictures of the flow cytometer (does not need separate labelling as the result of these are plotted on the columns) but increase at least the treatment name and add the number of “% of cells” on the white area of the flow result window/ in the squares/.

4. Figure 4-5 the same: add only two subfigure /page with (2-2-1 vertically arranged), enlarge, MTT as title and axis name only which now in the brackets

5. Figure 6-7: similar changes need like above.

Suggestion for the content change

1. Abstract:

Change the “Canagliflozin (Can) and dapagliflozin (Dap) possessed cytotoxic, antiproliferative and pro-apoptotic properties in the tested PC cell lines” sentence to “Canagliflozin and dapagliflozin possessed cytotoxic, antiproliferative and pro-apoptotic properties in the tested PC cell lines.”

Line 20: use “SGLT1 inhibitors” if both had these effects in PANC-1 cells in combination with Met OR specify which one had these effect.

Change the “The cytotoxic effects of Can and Dap in AsPC-1 cells, but not the cytotoxic effects of Can+Met and Dap+Met, involves PI3K and 21 JNK-dependent pathways.” to “Only the cytotoxic effects of SGLT2 inhibitors, but not the combination treatments, involved PI3K and JNK-dependent pathways in AsPC-1 cells.”

Change the “In conclusion, combination of an SGLT2 inhibitor with Met increased some of the in vitro anticancer effects of the compounds, in a PC cell type-specific way. Moreover, the cytotoxic effect of Dap and Can in AsPC-1 cells is dependent on PI3K and JNK pathways, but Met appears to act by a distinct mechanism.” to “In conclusion, combination treatments increased the anticancer effects in cell type dependent way in the two investigated cells. In PANC-1 (desctive what was the effect) and in as PC-1 (describe what was the effect). Additionally, we found that cytotoxic effect of SGLT2 inhibitors was dependent on PIK3 and JNK pathway, but Met appears to act by a distinct mechanism.”

Similar changes in the focus and structure need also in the result section and in the discussion as well.

2. Introduction:

Please cite this recently published paper (Foretz, M.; Guigas, B.; Viollet, B. Metformin: update on mechanisms of action and repurposing potential. Nat Rev Endocrinol 1298 2023, 19, 460-476, doi:10.1038/s41574-023-00833-4.) with a description of the updated mechanism of action of metformin's in line 45, before the sentence, which starts with “Importantly…”:

3. Please change the “Interestingly, “ to “Importantly, “ at the end of the line 55. It is interesting but not that much – if we think on that cancer need more energy, thus, promote glucose reabsorption and uptake etc. from everywhere - but more important because of your sentence there have already described why.….

4. Results:

In the 2.1 and 2.2 result sections always missing the “significantly” adjective, which is important. Please add in each places where appropriate “significant decrease” or “significantly decreased” or “significant, concentration-dependent decrease” etc.

Try to create logical blocks for description – not always useful the 1 by 1 methodology grouping for presentation. Present the most significant first and then the less important results even if you use the temporal order of the experiments.

5. Discussion:

The feeling of the reader is that there are not an exact conclusion of the sections in the discussion, just simply a listing what others have done. The discussion of the results of the glucose –uptake and migration assay is not completely explained for example, no exact conclusion (in lines 343-378).

In this section it is even more true that discuss first the most important result, combination treatments if this was the most important. You can outline how the experimental set-up was but then start the discussion with the strong, significant combination results and then bring those which less, then come which already published by others (i.e. the effects of the single) but needed to be able to plan the combination ones.
